# Resting heart rate is a population-level biomarker of cardiorespiratory fitness: The Fenland Study

**Tomas I. Gonzales**[1], **Justin Y. Jeon**[1,2], **Timothy Lindsay**[1], **Kate Westgate**[1], **Ignacio Perez-Pozuelo**[1], **Stefanie Hollidge**[1], **Katrien Wijndaele**[1], **Kirsten Rennie**[1], **Nita Forouhi**[1], **Simon Griffin**[1], **Nick Wareham**[1], **Soren Brage**[1]*

**1** MRC Epidemiology Unit, University of Cambridge, Cambridge, United Kingdom, **2** Department of Sport Industry Studies, Exercise Medicine Center for Diabetes and Cancer Patients (ICONS), Yonsei University, Seoul, Korea

* Soren.brage@mrc-epid.cam.ac.uk

## Abstract

### Introduction

Few large studies have evaluated the relationship between resting heart rate (RHR) and cardiorespiratory fitness. Here we examine cross-sectional and longitudinal relationships between RHR and fitness, explore factors that influence these relationships, and demonstrate the utility of RHR for remote population monitoring.

### Methods

In cross-sectional analyses (The UK Fenland Study: 5,722 women, 5,143 men, aged 29-65y), we measured RHR (beats per min, bpm) while seated, supine, and during sleep. Fitness was estimated as maximal oxygen consumption (ml·min⁻¹·kg⁻¹) from an exercise test. Associations between RHR and fitness were evaluated while adjusting for age, sex, adiposity, and physical activity. In longitudinal analyses (6,589 participant subsample), we reassessed RHR and fitness after a median of 6 years and evaluated the association between within-person change in RHR and fitness. During the coronavirus disease-2019 pandemic, we used a smartphone application to remotely and serially measure RHR (1,914 participant subsample, August 2020 to April 2021) and examined differences in RHR dynamics by prepandemic fitness level.

### Results

Mean RHR while seated, supine, and during sleep was 67, 64, and 57 bpm. Age-adjusted associations (beta coefficients) between RHR and fitness were -0.26, -0.29, and -0.21 ml·kg⁻¹·beat⁻¹ in women and -0.27, -0.31, and -0.19 ml·kg⁻¹·beat⁻¹ in men. Adjustment for adiposity and physical activity attenuated the RHR-to-fitness relationship by 10% and 50%, respectively. Longitudinally, a 1-bpm increase in supine RHR was associated with a 0.23 ml·min⁻¹·kg⁻¹ decrease in fitness. During the pandemic, RHR increased in those with low pre-pandemic fitness but was stable in others.

**Data Availability Statement:** The datasets generated and analysed during the current study are available at request via the MRC Epidemiology website (http://www.mrc-epid.cam.ac.uk/research/

data-sharing/). Authors were not precluded from accessing data in the study.

**Funding:** The authors were supported by the UK Medical Research Council (https://www.ukri.org/councils/mrc/) (N.W., grant number MC_UU_12015/1, MC_UU_00006/1), (T.I.G., S.B., K.Wi., S.H. grant number MC_UU_12015/3, , MC_UU_00006/4), (S.G., grant number MC_UU_12015/4, , MC_UU_00006/5), (N.G.F., grant number MC_UU_12015/5, , MC_UU_00006/3), and the NIHR Cambridge Biomedical Research Centre (K.We., S.B., N.G.F., and N.W., grant number IS-BRC-1215-20014). The funders had no role in study design, data collection and analysis, decision to publish, or preparation of the manuscript.

**Competing interests:** The authors have declared that no competing interests exist.

**Abbreviations:** RHR, Resting heart rate; UK, United Kingdom; VO$_2$max, Maximal oxygen consumption; TBM, Total-body mass; FFM, Fat-free mass; PAEE, Physical activity energy expenditure; MET, Metabolic equivalent; BMI, Body-mass index; FMI, Fat-mass index; COVID-19, Coronavirus disease-19; DEXA, Dual-energy X-ray absorptiometry.

## Conclusions

RHR is a valid population-level biomarker of cardiorespiratory fitness. Physical activity and adiposity attenuate the relationship between RHR and fitness.

## Introduction

Biomarkers can be broadly classified by their individual- and population-level applicability. Individual-level biomarkers are used in clinical environments for the diagnosis, prognosis, and treatment of disease. Population-level biomarkers are used in epidemiological and public health settings for screening, surveillance, and to monitor change in response to policy interventions. Whereas individual-level biomarkers need to be precise given potential clinical applications, population-level biomarkers need only be associated with the health construct they measure, as well as being easy to measure in field scenarios, low-cost, and scalable. The gold-standard biomarker of cardiorespiratory fitness (hereafter referred to as 'fitness') is maximal oxygen consumption (VO$_2$max). Higher VO$_2$max levels are associated with the lower incidence of type 2 diabetes, cardiovascular disease, cancer [1–3], and mortality [4]. VO$_2$max is not routinely measured in epidemiological and public health settings, however measurement of resting heart rate (RHR) could be a viable alternative at the population level [5]. RHR is inversely related to fitness, is relatively easy to measure, and has associations with health outcomes that are similar to those observed with fitness [6–8]. However, while these previous studies measure both fitness and RHR, they do not explore how behavioural health factors commonly measured in population research influence the RHR-to-fitness relationship.

The use of RHR as a population-level biomarker of fitness is scarce, partly for methodological reasons. RHR can vary depending on an individual's physiological state and posture at the time of measurement. The most common postures used in clinical environments are sitting upright during blood pressure measurement or lying supine during brief multi-lead electrocardiogram measurement. In free-living, wearable sensors conveniently measure RHR in other states of rest, particularly sleep. It is unknown whether differences in RHR measured in these ways alter the relationship with fitness. This is particularly problematic in population-based clinical research settings, where RHR could be measured in different ways across research centres and participant subgroups. It is also unclear whether the RHR-to-fitness relationship is affected by adiposity and physical activity, which have established impact on fitness [9]. Quantifying the influence these modifiable factors have on the RHR-to-fitness relationship would allow population-based researchers to understand how this relationship varies across population strata. Finally, although some studies have described the longitudinal relationship between RHR and fitness [5,10], there is uncertainty about how individual changes in RHR may reflect longer-term fitness changes in the general population.

In a large UK population-based study, we assess cross-sectional associations between different measures of RHR and fitness estimated from a submaximal treadmill test [11]. We explore how adiposity and physical activity alter the RHR-to-fitness relationship and determine whether longitudinal within-person change in RHR is associated with within-person change in fitness. We then demonstrate the use of remote RHR monitoring for population surveillance of fitness using a smartphone application during the coronavirus disease 2019 (COVID-19) pandemic.

## Methods

### Study design and participants

Participants born 1950–1975 were recruited between 2005 and 2015 from general practice lists around Cambridgeshire in England to the Fenland Study, a population-based cohort study set up to examine genetic and behavioural risk factors for metabolic disease in the general population [11]. The present study is an original analysis of data collected over the course of the Fenland Study. Exclusion criteria were prevalent diabetes, pregnancy or lactation, inability to walk unaided, psychosis or terminal illness. The study complies with the items listed in the Strengthening the Reporting of Observational Studies in Epidemiology (STROBE) guidelines, the Declaration of Helsinki, and was approved by the Health Research Authority NRES Committee East of England-Cambridge Central. All participants gave written informed consent. The Fenland Study has a dedicated Patient and Public Involvement panel, who provided input on the acceptability of the study protocols and how participant data confidentiality was ensured.

A total of 12,435 individuals participated in baseline assessments (response rate 27%). For the present analysis, data from 10,865 individuals (5,722 women, 5,143 men) were included after excluding participants on beta-blockers (n = 315) and those missing information on RHR (n = 28), fitness (n = 846), physical activity (n = 373), and adiposity (body-mass index, BMI; fat-mass index, FMI; n = 8). Compared to the main cohort sample, included participants were 0.4 years younger, 2% more physically active, had a 0.2 bpm lower RHR, were 1 cm taller, had 0.5 kg lower total-body mass, and 0.5% less bodyfat, but were similar across other characteristics (S1 Table in S1 File). In a subsample of 6,589 participants (available at time of analysis; 3,349 women, 3,240 men), RHR and fitness were re-assessed (2014–2020) after a median (interquartile range) of 6 (5–8) years of follow-up, allowing within-person longitudinal change in RHR and fitness to be examined. All participants were invited to be re-assessed, unless they had died or asked not to be contacted again. An additional subsample of 1,914 participants (1,038 women, 876 men) participated in remote heart rate monitoring from August 2020 to April 2021 during the coronavirus disease 2019 (COVID-19) pandemic (the Fenland COVID-19 substudy). All participants who had attended a Fenland Study visit between 2005 and 2015, owning a smartphone, and living in the UK were invited to take part in the Fenland COVID-19 substudy. This information is summarised in S1 Fig in S1 File.

### Resting heart rate measurement

Participants arrived at a clinical testing facility after an overnight fast to complete baseline and follow-up assessments and questionnaires. Resting pulse rate was measured in a seated position while blood pressure was assessed three times at 1-minute intervals (Omron 705CP-II, OMRON Healthcare Europe, Hoofddorp, Netherlands). Seated RHR was computed as the mean of the three pulse rate values. At least one hour after arrival, RHR was measured with the participant at rest in a supine position using a combined heart rate and movement sensor (Actiheart, CamNtech, Papworth, UK) attached to the chest at the base of the sternum by two electrocardiogram electrodes. Heart rate was recorded for 6 minutes and RHR was calculated as the mean heart rate measured during the last 3 minutes. Following the clinical visit, participants wore the same combined heart rate and movement sensor continuously for 6 days and nights during free-living, recording at 60-second intervals. Habitual sleeping heart rate was estimated from these data by first deriving a robust daily minimum heart rate as the thirtieth lowest minute-by-minute heart rate reading during each 24-hour period, and then averaging across days [12]. For longitudinal analyses, supine RHR was measured using a 15-min rest

protocol at follow-up. Only minutes 4 to 6 was used here to match the baseline design for derivation of change in RHR.

We used a custom-designed smartphone application (designed by Huma Therapeutics Ltd.) to capture weekly remote measurements of RHR during the COVID-19 pandemic. The smartphone application was designed specifically for this aspect of the study and is not commercially available. Participants were asked to complete four measurement modules three times a week: RHR using their smartphone, oxygen saturation using a pulse oximeter, body temperature, and COVID symptom recording. For reasons of practicality and to control diurnal variation, participants were asked to take all measurements first thing in the morning after awaking. Participants were asked to measure their RHR by placing their finger over the camera on their smartphone. The measurement took approximately 60 seconds to provide a measure of RHR [13]. Further instructions on how to take these measurements in visual or video form are provided on the study website [14]. Smartphone camera-based photoplethysmography methods for RHR measurement have been validated previously [15]. RHR could also be measured using the pulse oximeter, however this was not the primary approach used for this analysis.

## Cardiorespiratory fitness measurement

Cardiorespiratory fitness, expressed as $VO_2max$, was estimated using heart rate response to a submaximal treadmill test [16]. Participants who self-reported having a heart condition or chest pain were examined by a study nurse to determine whether treadmill testing could be conducted safely. Participants exercised while treadmill speed and grade increased across four stages of level walking, inclined walking, and level running. The first stage consisted of walking at 3.2 km·h$^{-1}$ at 0% incline for 3 min. The second stage consisted of walking while treadmill speed increased from 3.2 to 5.2 km·h$^{-1}$ at 0% incline for 6 min. The third stage consisted of walking at 5.2 km·h$^{-1}$ while incline increased from 0 to 6% for 3 min, and then walking while treadmill speed increased from 5.2 to 5.8 km·h$^{-1}$ and incline from 6 to 10.2% for 3 min. The fourth consisted of running while treadmill speed increased from 5.8 to 9.0 km·h$^{-1}$ and incline decreased to 0% for 1 min, and then running while treadmill speed increased from 9.0 to 12.6 km·h$^{-1}$ and at 0% incline for 4.5 min. Testing was terminated if one of the following criteria were met: 1) the participant wanted to stop, 2) the participant reached 90% of age-predicted maximal heart rate (208–0.7 x age); or 3) the participant had exercised above 80% of age-predicted maximal heart rate for >2 minutes. Predicted workload (physical activity intensity, in J·min$^{-1}$·kg$^{-1}$) during the treadmill protocol was regressed against heart rate to define the individual's submaximal relationship between heart rate and physical activity intensity. The heart rate-to-physical activity intensity relationship was then extrapolated to age-predicted maximal heart rate to predict maximal work capacity. The resulting work capacity was converted to $VO_2max$ by adding an estimate of resting metabolic rate and dividing by the energetic equivalent of oxygen [17]. $VO_2max$ estimates were expressed in both ml $O_2$·min$^{-1}$·kg$^{-1}$ total-body mass and as ml $O_2$·min$^{-1}$·kg$^{-1}$ fat-free mass. This $VO_2max$ estimation procedure has been validated against directly measured $VO_2max$ in a subsample of Fenland Study participants (43 women; 42 men), demonstrating low bias and high correlation with directly measured $VO_2max$ (r = 0.70) [18,19]. For longitudinal analyses, fitness was assessed using the same methodology that was used at baseline.

## Physical activity measurement

Physical activity was objectively measured over 6 days using a combined heart rate and movement sensor (Actiheart, CamNtech, Papworth, UK) with individual calibration of the heart

rate-to-physical activity intensity relationship performed using data from the treadmill test [11]. Free-living heart rate data was pre-processed [20] and modelled using a branched equation framework [21] to estimate physical activity intensity time-series, which was then summarised over time as daily physical activity energy expenditure (PAEE) (kJ·day$^{-1}$·kg$^{-1}$). This method has been validated against gold-standard measures of energy expenditure in laboratory and free-living conditions [22–24]. Intensity was expressed in metabolic equivalents (METs), using 1 MET = 71 J·min$^{-1}$·kg$^{-1}$ (~3.5 ml O$_2$·min$^{-1}$·kg$^{-1}$), and summarised as moderate-to-vigorous physical activity for intensity greater than 3.0 gross METs (activity intensity above 142 J·min$^{-1}$·kg$^{-1}$). In the present analyses, the proportional PAEE accumulated in moderate and vigorous physical activity was used in combination with total PAEE to account for both total physical activity and higher intensity activity.

## Sociodemographic and anthropometric characteristics

Ethnicity (White, non-White), smoking status (never, former, current), and alcohol intake (units/week) were determined using a self-administered questionnaire. Anthropometric measures were collected by trained personnel. Weight was measured with a calibrated electronic scale (TANITA model BC-418 MA; Tanita, Tokyo, Japan) and height was assessed with a calibrated stadiometer (SECA 240; Seca, Birmingham, United Kingdom). Adiposity was assessed using dual-energy X-ray absorptiometry (DEXA; GE Lunar Prodigy Advanced fan beam scanner, GE Healthcare, Bedford, United Kingdom) deriving fat, lean and bone mass estimates across body regions.

## Statistical analyses

Interrelationships of RHR measures were examined by linear regression. We used sex-stratified regression models to examine cross-sectional associations between RHR and fitness while adjusting for confounding or explanatory factors. Five progressively adjusted models were used. Model 1 adjusted for age only. Model 2 additionally adjusted for demographic and lifestyle factors (ethnicity, smoking status, alcohol intake). Model 3 additionally adjusted for BMI. Model 4 added adjustment for PAEE, and Model 5 further adjusted for moderate-to-vigorous intensity activity (as a fraction of total PAEE). When fitness was expressed relative to fat-free mass (FFM) instead of total-body mass, fat mass index (fat-mass divided by height squared) was used to adjust for adiposity instead of BMI. Subgroup analyses were performed as follows: Analyses were stratified by groups of age (less than 50y; 50-59y; 60y and greater), BMI (normal weight, BMI 18.5–25; overweight, BMI 25–30; obese, BMI above 30 kg·m$^{-2}$), and PAEE level (<40, 40–60, >60 kJ·day$^{-1}$·kg$^{-1}$). As a sensitivity analysis, we also examined the relationship between RHR and fitness in the subsample with direct VO$_2$max measurements [18], adjusting for age and BMI. Descriptive statistics were reported as means and standard deviations, unless specified otherwise. For longitudinal analyses of the subsample with repeat measures of RHR and fitness, associations between within-person change in RHR and fitness were adjusted for baseline age, sex, RHR, fitness, and age at follow-up.

   Participants with missing continuous covariate data were excluded from the analyses, whereas participants with missing categorical covariate data were coded as a separate category and included in the analyses.

   For analysis of smartphone RHR data, we used a random effects linear spline regression model to examine heart rate dynamics over time, with spline knots located at the dates for the 2$^{nd}$ and 3$^{rd}$ UK national lockdowns (5 November 2020 and 6 January 2021). Analyses were conducted on the pooled sample and also stratified by sex-specific fitness tertiles (low, mid, high), where within participant fitness was computed as the average of VO$_2$max estimates

from the initial and follow-up submaximal treadmill test. Sensitivity analyses were also conducted with alternative stratification for pre-pandemic RHR tertiles and BMI groups (normal weight, overweight, and obese).

Statistical analyses were performed with STATA (Version 14.2; StataCorp, College Station, TX); a p-value of 0.05 or less was considered statistically significant. P-values for linear spline regression modelling were adjusted by using robust standard errors, allowing for clustering of serial RHR measurements within participants.

## Results

Participant characteristics are summarised in Table 1, stratified by sex and supine RHR categories. At baseline, mean ± standard deviation RHR while seated, supine, and during sleep was 67.6 ± 9.8, 63.5 ± 8.9, and 56.9 ± 6.9 bpm, respectively. Those with higher seated, supine, and sleeping RHR had higher BMI and body fat levels, lower estimated $VO_2max$, and were less physically active. Median (interquartile range) treadmill duration was 15 (14–16) minutes for women and 16 (15–17) minutes for men. On average, RHR was 3 bpm higher and $VO_2max$ estimates were 7.7 ml $O_2 \cdot min^{-1} \cdot kg^{-1}$ total body-mass lower in women compared to men. Participant characteristics stratified by fitness tertiles are provided in S2 Table in S1 File. Correlations (Pearson r) of RHR values between measurement approaches ranged from 0.65 to 0.81 (S3 Table in S1 File).

Cross-sectional associations between RHR and estimated $VO_2max$ per kg total-body mass (Table 2 and Fig 1) and per kg fat-free mass (S4 Table in S1 File and S2 Fig in S1 File) were explored using a series of sequentially-adjusted regression models. In models with only age adjustment (Model 1), RHR was significantly associated with $VO_2max$ estimates in both women and men, irrespective of which RHR measure was used. Associations were similar after adjustment for ethnicity, smoking, and alcohol use (Model 2). Further adjustment by BMI (Model 3) attenuated associations by about 10% for $VO_2max$ per kg total-body mass. Fat mass index (FMI) adjustment in models of estimated $VO_2max$ per kg fat-free mass resulted in stronger associations, particularly for sleeping RHR which had beta coefficients 60% larger in magnitude. Adjustment for PAEE (Model 4) attenuated the RHR-to-fitness relationship by 30–40% irrespective of fitness normalisation, with 5–15% additional attenuation when proportion of PAEE expended at moderate and vigorous intensity was accounted for (Model 5). Associations between RHR and estimated fitness were similar across all RHR modalities, especially in maximally adjusted models, but BMI and physical activity attenuated more of the relationship for sleeping heart rate in women (30% and 40%, respectively). We also analysed the association between RHR and estimated fitness across age, BMI, and PAEE strata separately. Associations in these subgroups were similar in strength to pooled associations and remained statistically significant. Associations in the subsample of participants with direct $VO_2max$ measurements are reported in S5 Table in S1 File. Parameter estimates reported above were within the confidence intervals in this analysis.

In longitudinal analyses, mean levels of RHR and estimated $VO_2max$ were similar to baseline values but with diverse individual change over the duration of the follow-up period. The 5th to the 95th percentiles of change were -11.4 to 9.2 bpm and -10.1 to 10.8 ml $O_2 \cdot min^{-1} \cdot kg^{-1}$, respectively. Correlations between baseline and follow-up measures were high for both RHR (r = 0.70) and estimated $VO_2max$ (r = 0.64). In longitudinal association analyses, adjusting for baseline age, sex, RHR, fitness, and age at follow-up, each 1-bpm increase in supine RHR was associated with a 0.23 (95%CI 0.20; 0.25) ml $O_2 \cdot min^{-1} \cdot kg^{-1}$ decline in estimated $VO_2max$; the correlation between them was r = -0.20. Sex-stratified beta coefficients were -0.21 (95%CI

**Table 1. Baseline participant characteristics in women and men stratified by supine resting heart rate.** The Fenland Study.

| Women | | | | | |
|---|---|---|---|---|---|
| | <60 bpm n = 1,724 | 60–69 bpm n = 2,623 | 70–79 bpm n = 1,162 | ≥80 bpm n = 213 | Total N = 5,722 |
| | Mean ± SD | Mean ± SD | Mean ± SD | Mean ± SD | Mean ± SD |
| RHR | | | | | |
| Seated (bpm) | 59.5 ± 6.0 | 67.9 ± 5.8 | 75.6 ± 6.1 | 84.4 ± 8.6 | 67.5 ± 9.0 |
| Sleeping (bpm) | 52.8 ± 5.0 | 58.9 ± 4.8 | 63.7 ± 5.3 | 68.0 ± 6.7 | 58.4 ± 6.6 |
| Cardiorespiratory fitness | | | | | |
| VO$_2$max per kg BM | 39.2 ± 9.3 | 35.8 ± 8.1 | 33.5 ± 7.8 | 30.7 ± 7.7 | 36.2 ± 8.7 |
| VO$_2$max per kg FFM | 60.7 ± 13.0 | 57.3 ± 12.3 | 55.2 ± 12.5 | 51.7 ± 13.0 | 57.7 ± 12.8 |
| Anthropometrics | | | | | |
| Height (m) | 1.65 ± 0.06 | 1.64 ± 0.06 | 1.63 ± 0.06 | 162.5 ± 5.9 | 1.64 ± 0.06 |
| Body mass (kg) | 69.1 ± 12.5 | 70.4 ± 13.4 | 72.0 ± 15.3 | 73.3 ± 16.6 | 70.4 ± 13.7 |
| BMI (kg·m$^{-2}$) | 25.4 ± 4.4 | 26.2 ± 4.8 | 27.0 ± 5.4 | 27.7 ± 5.8 | 26.1 ± 4.9 |
| FMI (kg·m$^{-2}$) | 9.2 ± 3.7 | 10.1 ± 3.8 | 10.9 ± 4.1 | 11.5 ± 4.3 | 10.0 ± 3.9 |
| Percent body fat (%) | 35.2 ± 8.3 | 37.3 ± 7.7 | 39.0 ± 7.5 | 40.2 ± 7.7 | 37.1 ± 8.0 |
| Physical Activity | | | | | |
| PAEE (kJ·day$^{-1}$·kg$^{-1}$) | 59.4 ± 21.3 | 49.3 ± 18.2 | 42.7 ± 16.4 | 36.2 ± 14.1 | 50.5 ± 19.8 |
| MVPA (min·day$^{-1}$) | 115.4 ± 78.5 | 82.0 ± 61.2 | 63.0 ± 50.9 | 43.4 ± 39.0 | 86.8 ± 67.8 |
| MVPA (kJ·day$^{-1}$·kg$^{-1}$) | 24.1 ± 17.5 | 16.3 ± 12.8 | 12.2 ± 10.4 | 8.3 ± 7.7 | 17.6 ± 14.6 |
| Age (years) | 48.4 ± 7.5 | 48.2 ± 7.3 | 47.8 ± 7.3 | 47.6 ± 7.0 | 48.2 ± 7.4 |
| | Count (%) | Count (%) | Count (%) | Count (%) | Count (%) |
| Ethnicity | | | | | |
| White | 1,619 (28.3) | 2,425 (42.4) | 1,074 (18.8) | 188(3.3) | 5,306 (92.7) |
| Non-White | 106 (1.9) | 197 (3.4) | 88 (1.5) | 25(0.4) | 416 (7.3) |
| Smoker Status | | | | | |
| Never smoked | 894 (15.6) | 1497 (26.2) | 684 (12.0) | 141(2.5) | 3,216 (56.2) |
| Ex-smoker | 605(10.6) | 819 (14.3) | 355 (6.2) | 51(0.9) | 1,830 (32.0) |
| Current smoker | 196 (3.4) | 280 (5.0) | 112 (2.0) | 20(0.3) | 608 (10.6) |
| Unknown | 30 (0.5) | 26 (0.5) | 11 (0.2) | 1(0.0) | 68 (1.2) |
| Alcohol Consumption | | | | | |
| <1/week | 611 (10.7) | 1,016 (17.7) | 501 (8.8) | 96(1.7) | 2,224 (38.9) |
| 1-4/week | 865 (15.1) | 1,264 (22.1) | 491 (8.6) | 82(1.4) | 2,702 (47.2) |
| Almost daily | 218 (3.8) | 297 (5.2) | 150 (2.6) | 30(0.5) | 695 (12.2) |
| Unknown | 30 (0.5) | 46 (0.8) | 20 (0.3) | 5(0.1) | 101 (1.8) |
| Men | | | | | |
| | <60 bpm n = 2,385 | 60–69 bpm n = 1,925 | 70–79 bpm n = 696 | ≥80 bpm n = 137 | Total N = 5,143 |
| | Mean ± SD | Mean ± SD | Mean ± SD | Mean ± SD | Mean ± SD |
| RHR | | | | | |
| Seated (bpm) | 57.9 ± 6.7 | 67.2 ± 6.1 | 75.3 ± 6.7 | 85.4 ± 8.8 | 64.5 ± 9.6 |
| Sleeping (bpm) | 50.9 ± 5.1 | 57.3 ± 5.0 | 61.9 ± 5.2 | 65.1 ± 6.6 | 55.2 ± 6.7 |
| Cardiorespiratory fitness | | | | | |
| VO$_2$max per kg BM | 46.4 ± 9.2 | 42.7 ± 8.6 | 40.4 ± 9.0 | 36.6 ± 8.6 | 43.9 ± 9.3 |
| VO$_2$max per kg FFM | 62.6 ± 12.0 | 59.6 ± 11.8 | 57.5 ± 12.3 | 53.7 ± 12.6 | 60.5 ± 12.2 |
| Anthropometrics | | | | | |
| Height (m) | 1.78 ± 0.07 | 1.77 ± 0.07 | 1.77 ± 0.07 | 176.7 ± 6.8 | 1.78 ± 0.07 |
| Body mass (kg) | 84.6 ± 12.6 | 86.2 ± 14.15 | 87.7 ± 15.1 | 91.1 ± 16.5 | 58.8 ± 13.7 |
| BMI (kg·m$^{-2}$) | 26.6 ± 3.6 | 27.4 ± 4.0 | 28.1 ± 4.4 | 29.1 ± 4.8 | 27.2 ± 4.0 |

*(Continued)*

**Table 1.** (Continued)

| | Women | | | | |
|---|---|---|---|---|---|
| FMI (kg·m$^{-2}$) | 7.0 ± 2.7 | 7.9 ± 2.8 | 8.5 ± 3.0 | 9.5 ± 3.2 | 7.6 ± 2.9 |
| Percent body fat (%) | 25.7 ± 7.1 | 28.1 ± 6.5 | 29.6 ± 6.6 | 31.7 ± 6.5 | 27.3 ± 7.0 |
| Physical Activity | | | | | |
| PAEE (kJ·day$^{-1}$·kg$^{-1}$) | 66.1 ± 23.5 | 55.9 ± 21.4 | 48.1 ± 20.9 | 39.0 ± 15.4 | 59.1 ± 23.4 |
| MVPA (min·day$^{-1}$) | 147.9 ± 88.6 | 114.0 ± 77.0 | 90.7 ± 71.6 | 61.9 ± 45.9 | 125.2 ± 84.6 |
| MVPA (kJ·day$^{-1}$·kg$^{-1}$) | 33.0 ± 20.4 | 23.7 ± 16.6 | 18.4 ± 15.2 | 12.3 ± 9.7 | 27.0 ± 19.1 |
| Age (years) | 48.0 ± 7.5 | 48.2 ± 7.5 | 48.8 ± 7.7 | 49.2 ± 8.0 | 48.2 ± 7.5 |
| | Count (%) | Count (%) | Count (%) | Count (%) | Count (%) |
| Ethnicity | | | | | |
| White | 2,227 (43.3) | 1,776 (34.5) | 627 (12.2) | 126 (2.4) | 4,756 (92.5) |
| Non-White | 158 (3.1) | 149 (2.9) | 69 (1.3) | 11 (0.2) | 387 (7.5) |
| Smoker Status | | | | | |
| Never smoked | 1,228 (23.9) | 1,028 (20.0) | 346 (6.7) | 76 (1.5) | 2,678 (52.1) |
| Ex-smoker | 831 (16.2) | 620 (12.1) | 239 (4.7) | 48 (0.9) | 1,738 (33.8) |
| Current smoker | 303 (5.9) | 255 (5.0) | 102 (2.0) | 12 (0.2) | 672 (13.1) |
| Unknown | 23 (0.4) | 22 (0.4) | 9 (0.2) | 1 (0.0) | 55 (1.1) |
| Alcohol Consumption | | | | | |
| <1/week | 563 (11.0) | 518 (10.1) | 193 (3.8) | 38 (0.7) | 1,312 (25.5) |
| 1-4/week | 1,351 (26.3) | 1,005 (19.5) | 342 (6.7) | 72 (1.4) | 2,770 (53.9) |
| Almost daily | 446 (8.7) | 368 (7.2) | 152 (3.0) | 25 (0.5) | 991 (19.3) |
| Unknown | 25 (0.5) | 34 (0.7) | 9 (0.2) | 2 (0.0) | 70 (1.4) |

SD: Standard deviation, RHR: Resting heart rate, bpm: Beat per minute, BMI: Body mass index, FMI: Fat mass index, PAEE: Physical activity energy expenditure. VO$_2$max: Estimated maximal oxygen consumption, MVPA: Moderate to vigorous physical activity, BM: Total-body mass, FFM: Fat free mass. Stratification by supine RHR categories.

-0.24; -0.17) and -0.25 (95%CI -0.28; -0.21) ml O$_2$·min$^{-1}$·kg$^{-1}$ per 1-bpm increase in RHR in women and men, respectively (Fig 2).

Having established cross-sectional and longitudinal associations between RHR and fitness, we demonstrate an application for population surveillance by using a smartphone application to capture weekly remote measurements of RHR during the COVID-19 pandemic. RHR was stable prior to the 2nd UK national lockdown (0.0037 bpm/week change, 95%CI -0.023, 0.030), increasing after the 2nd UK national lockdown (0.040 bpm/week increase, 95%CI 0.017, 0.061), and increasing from the 3rd national lockdown onwards (0.025 bpm/week increase, 95%CI 0.0076, 0.042). We examined how these trends differed by sex-specific fitness tertiles (low, mid, and high), established by exercise testing prior to the COVID-19 pandemic (Fig 3). There was marked separation of RHR levels by fitness tertiles. In women, RHR was stable in all fitness groups prior to and during the 2nd UK national lockdown, however from the 3rd lockdown onwards, RHR was increasing in the low fitness group only (0.095 bpm/week increase, 95%CI 0.040, 0.15). In men, RHR was stable in all fitness groups prior to the 2nd UK national lockdown, increasing in the low fitness group only during the 2nd lockdown (0.13 bpm/week increase, 95%CI 0.070, 0.18), and stable in all groups from the 3rd UK national lockdown onwards. We also examined RHR trajectories stratified by pre-pandemic RHR tertiles (S3 Fig in S1 File) and BMI groups (S4 Fig in S1 File). In general, trends were similar to those observed when stratified by fitness level.

**Table 2. Association between resting heart rate and estimated maximal oxygen consumption expressed per kg total-body mass.** The Fenland Study.

| | | Seated RHR | | Supine RHR | | Sleeping RHR | |
|---|---|---|---|---|---|---|---|
| | | **Men** | **Women** | **Men** | **Women** | **Men** | **Women** |
| | | Total sample | | | | | |
| Model 1 | | -0.27 (-0.29, -0.24) | -0.25 (-0.27, -0.22) | -0.33 (-0.36, -0.30) | -0.31 (-0.33, -0.28) | -0.28 (-0.32, -0.24) | -0.27 (-0.31, -0.24) |
| Model 2 | | -0.27 (-0.29, -0.24) | -0.25 (-0.27, -0.22) | -0.32 (-0.35, -0.29) | -0.30 (-0.33, -0.28) | -0.31 (-0.35, -0.28) | -0.31 (-0.35, -0.28) |
| Model 3 | | -0.23 (-0.26, -0.21) | -0.19 (-0.22, -0.17) | -0.29 (-0.32, -0.26) | -0.29 (-0.32, -0.26) | -0.26 (-0.30, -0.23) | -0.22 (-0.25, -0.19) |
| Model 4 | | -0.16 (-0.18, -0.14) | -0.13 (-0.15, -0.11) | -0.18 (-0.21, -0.16) | -0.17 (-0.19, -0.14) | -0.17 (-0.21, -0.13) | -0.14 (-0.17, -0.11) |
| Model 5 | | -0.14 (-0.16, -0.12) | -0.12 (-0.14, -0.10) | -0.16 (-0.18, -0.13) | -0.15 (-0.17, -0.12) | -0.15 (-0.18, -0.11) | -0.13 (-0.16, -0.10) |
| | | Age stratified | | | | | |
| Model 2 | <50 years | -0.26 (-0.30, -0.22) | -0.25 (-0.29, -0.21) | -0.34 (-0.38, -0.29) | -0.29 (-0.34, -0.25) | -0.35 (-0.41, -0.29) | -0.30 (-0.35, -0.24) |
| | 50–60 years | -0.26 (-0.30, -0.23) | -0.25 (-0.28, -0.21) | -0.30 (-0.35, -0.26) | -0.31 (-0.35, -0.27) | -0.28 (-0.34, -0.22) | -0.32 (-0.37, -0.27) |
| | >60 years | -0.28 (-0.33, -0.22) | -0.24 (-0.30, -0.19) | -0.33 (-0.39, -0.27) | -0.32 (-0.38, -0.26) | -0.31 (-0.40, -0.23) | -0.33 (-0.41, -0.25) |
| Model 3 | <50 years | -0.22 (-0.26, -0.18) | -0.20 (-0.23, -0.16) | -0.30 (-0.34, -0.26) | -0.25 (-0.29, -0.21) | -0.29 (-0.35, -0.23) | -0.21 (-0.26, -0.16) |
| | 50–60 years | -0.23 (-0.27, -0.19) | -0.19 (-0.22, -0.15) | -0.27 (-0.31, -0.23) | -0.25 (-0.29, -0.22) | -0.23 (-0.29, -0.17) | -0.22 (-0.27, -0.17) |
| | >60 years | -0.25 (-0.31, -0.19) | -0.21 (-0.27, -0.16) | -0.30 (-0.36, -0.24) | -0.29 (-0.35, -0.23) | -0.28 (-0.36, -0.20) | -0.27 (-0.34, -0.19) |
| Model 5 | <50 years | -0.12 (-0.16, -0.08) | -0.10 (-0.14, -0.07) | -0.15 (-0.19, -0.11) | -0.11 (-0.15, -0.07) | -0.14 (-0.20, -0.09) | -0.10 (-0.15, -0.05) |
| | 50–60 years | -0.15 (-0.18, -0.11) | -0.11 (-0.14, -0.08) | -0.15 (-0.19, -0.11) | -0.14 (-0.18, -0.10) | -0.13 (-0.19, -0.08) | -0.12 (-0.17, -0.07) |
| | >60 years | -0.16 (-0.21, -0.11) | -0.17 (-0.22, -0.12) | -0.17 (-0.24, -0.11) | -0.22 (-0.28, -0.16) | -0.17 (-0.25, -0.10) | -0.22 (-0.29, -0.15) |
| | | BMI stratified | | | | | |
| Model 3 | <25 kg/m$^2$ | -0.30 (-0.35, -0.26) | -0.24 (-0.27, -0.20) | -0.35 (-0.40, -0.30) | -0.32 (-0.36, -0.28) | -0.34 (-0.41, -0.27) | -0.31 (-0.36, -0.26) |
| | 25–30 kg/m$^2$ | -0.21 (-0.24, -0.17) | -0.18 (-0.22, -0.14) | -0.27 (-0.31, -0.24) | -0.22 (-0.26, -0.17) | -0.24 (-0.30, -0.19) | -0.17 (-0.22, -0.11) |
| | >30 kg/m$^2$ | -0.18 (-0.23, -0.13) | -0.11 (-0.15, -0.06) | -0.22 (-0.28, -0.17) | -0.17 (-0.21, -0.12) | -0.19 (-0.26, -0.11) | -0.13 (-0.19, -0.07) |
| Model 5 | <25 kg/m$^2$ | -0.18 (-0.23, -0.14) | -0.13 (-0.17, -0.10) | -0.18 (-0.23, -0.13) | -0.17 (-0.21, -0.14) | -0.18 (-0.24, -0.11) | -0.18 (-0.23, -0.14) |
| | 25–30 kg/m$^2$ | -0.13 (-0.16, -0.09) | -0.13 (-0.17, -0.09) | -0.15 (-0.19, -0.12) | -0.13 (-0.17, -0.08) | -0.15 (-0.20, -0.10) | -0.09 (-0.14, -0.04) |
| | >30 kg/m$^2$ | -0.10 (-0.15, -0.05) | -0.08 (-0.12, -0.04) | -0.12 (-0.17, -0.06) | -0.11 (-0.16, -0.07) | -0.08 (-0.16, -0.01) | -0.08 (-0.14, -0.02) |
| | | PAEE stratified | | | | | |
| Model 5 | <40 kJ/day/kg | -0.13 (-018, -0.08) | -0.14 (-0.18, -0.11) | -0.18 (-0.23, -0.13) | -0.18 (-0.22, -0.14) | -0.11 (-0.19, -0.04) | -0.13 (-0.18, -0.08) |
| | 40–60 kJ/day/kg | -0.15 (-0.19, -0.11) | -0.10 (-0.13, -0.06) | -0.17 (-0.22, -0.13) | -0.12 (-0.16, -0.08) | -0.16 (-0.21, -0.10) | -0.13 (-0.17, -0.08) |
| | >60 kJ/day/kg | -0.13 (-0.17, -0.09) | -0.10 (-0.15, -0.06) | -0.12 (-0.16, -0.07) | -0.12 (-0.17, -0.07) | -0.16 (-0.22, -0.11) | -0.14 (-0.20, -0.08) |

Reported values are beta coefficients (95%CI) for the difference in estimated fitness (dependent variable) per a 1-bpm difference in resting heart rate (independent variable).

Model 1: Age-adjusted.

Model 2: Model 1 + ethnicity, smoking and alcohol adjusted.

Model 3: Model 2 + body mass index (BMI) adjusted.

Model 4: Model 3 + physical activity energy expenditure (PAEE) adjusted.

Model 5: Model 4 + moderate-vigorous intensity PAEE adjusted.

## Discussion

We have documented strong interrelationships between measures of RHR (when seated, lying supine, and during sleep) and investigated their relationship with cardiorespiratory fitness (estimated maximal oxygen consumption; VO$_2$max) in a large population-based study of UK adults. Cross-sectional analyses showed inverse associations between RHR and fitness that persisted across different RHR measurement modalities and fitness normalisation conventions (by total-body mass or by fat-free mass). Part of the association between RHR and fitness was explained by adiposity, but a greater proportion was explained by physical activity. In longitudinal analyses, within-person change in RHR was associated with within-person change in fitness, similar in magnitude to the relationship observed cross-sectionally. We demonstrate an

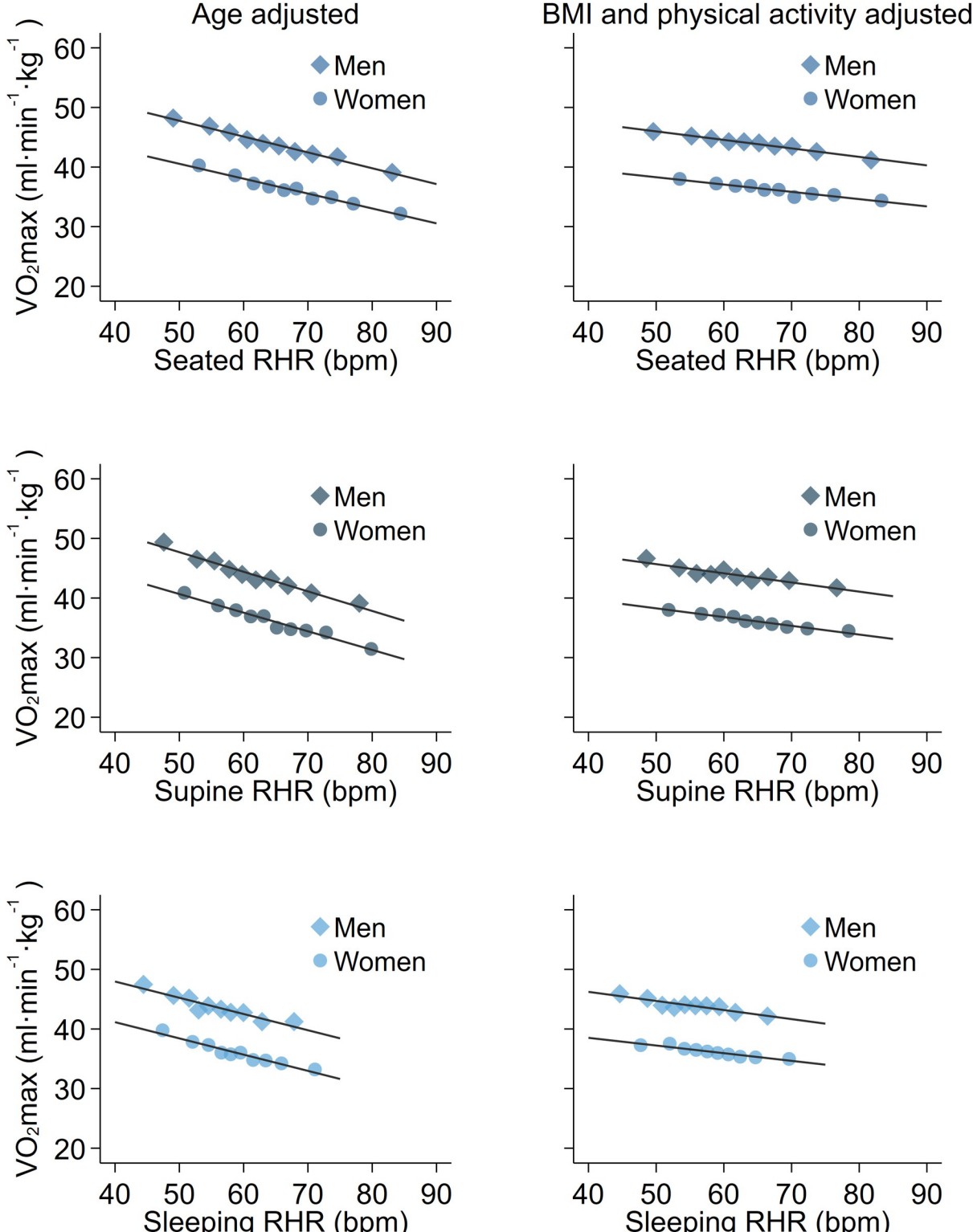

**Fig 1.** Associations between resting heart rate and estimated maximal oxygen consumption expressed per kg total-body mass, stratified by sex and adjusted for age (left column panels) or age, ethnicity, smoking, alcohol consumption, body mass index, physical activity energy expenditure (PAEE), moderate-vigorous PAEE (right column panels). Top: Seated resting heart rate. Middle: Supine resting heart rate. Bottom: Sleeping resting heart rate. The Fenland Study (n = 10,865). Each point represents 5% of data in the binscatter plots. r values are sex-stratified partial correlation coefficients between resting heart rate and estimated maximal oxygen consumption, adjusted for covariates listed above. P-values for all partial correlation coefficients are less than 0.0001.

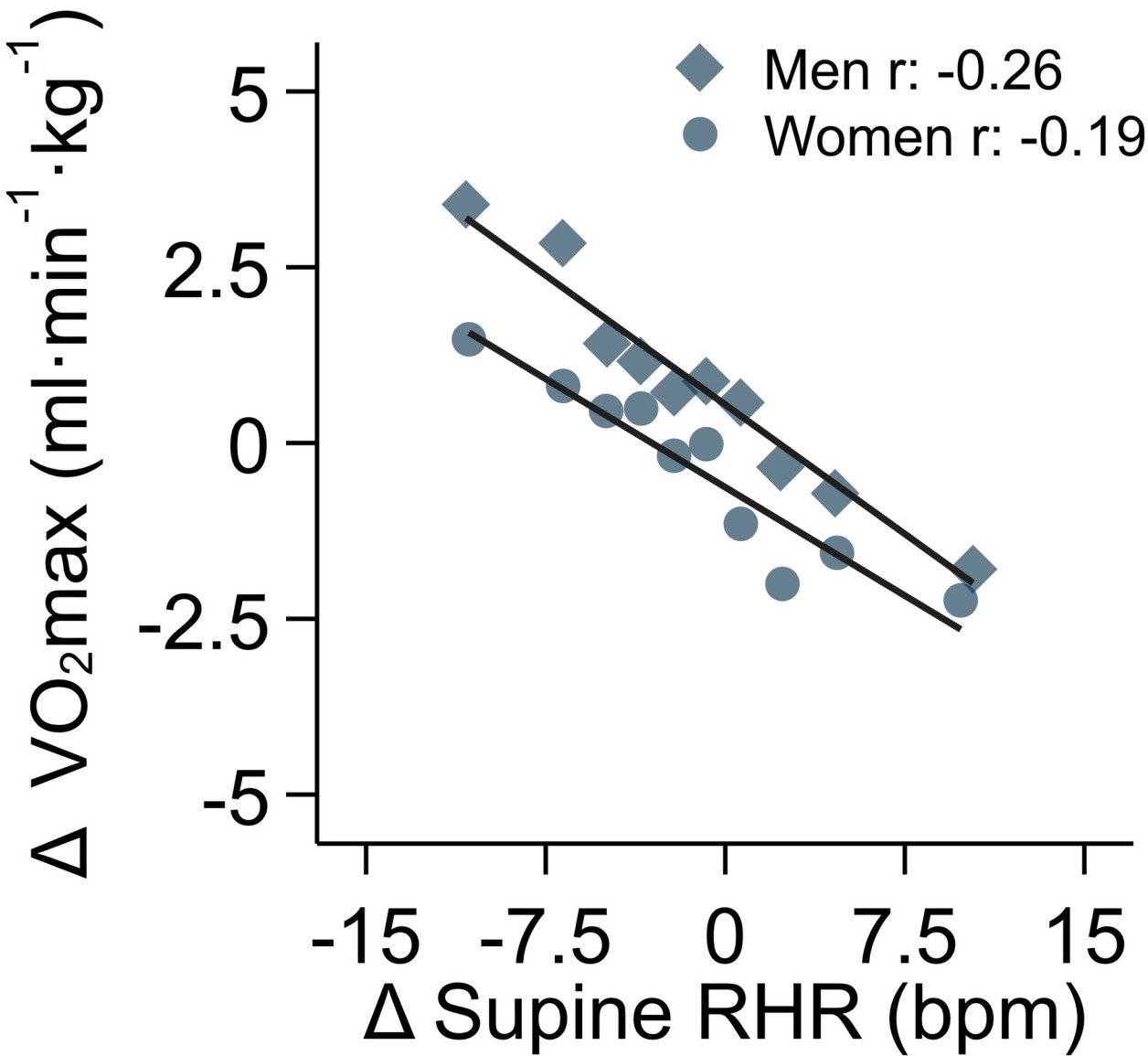

**Fig 2. Association between 6-year change in supine resting heart rate and change in estimated maximal oxygen consumption, stratified by sex.** Models were adjusted for follow-up time and baseline values of age, sex, RHR, and estimated $VO_2$max. Longitudinal subsample, the Fenland Study (n = 6,589). Each point represents 5% of the data in the binscatter plot. R values are sex-stratified partial correlation coefficients between change in supine resting heart rate and change in estimated maximal oxygen consumption, adjusted for covariates listed above. P-values for all partial correlation coefficients are less than 0.0001.

application for population monitoring of RHR by remotely capturing weekly RHR measurements with smartphones during the COVID-19 pandemic, showing differential trajectories of RHR during periods when opportunities for exercise were restricted. RHR may therefore be used as a feasible population-level biomarker of fitness, and changes in factors determining fitness are paralleled by those that influence RHR.

This is the first study to examine associations between multiple measures of RHR and fitness while also describing the effect of adjusting for objectively measured adiposity and physical activity in a large population cohort of men and women. Adjusting for BMI attenuated the association between RHR and fitness scale by total-body mass. However, the association

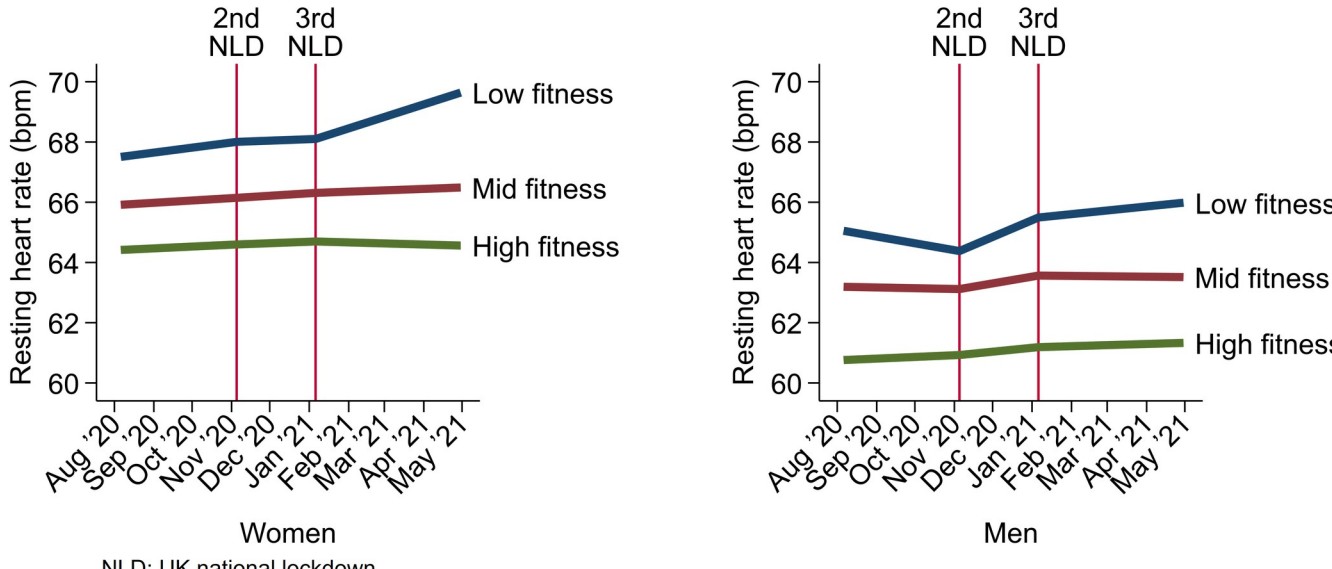

**Fig 3. Linear spline regression modelling of RHR during the COVID-19 pandemic, stratified by sex and pre-pandemic estimated cardiorespiratory fitness level.**

between RHR and fitness scaled by fat-free mass was stronger when adjusted by DEXA-measured adiposity, which may be explained by a better characterisation of differences in blood volume by fitness level [25]. We have shown that activity levels in this cohort are similar to those reported in national UK surveys [11,26], suggesting that fitness, as the capacity to undertake physical activity, may also be similar to national levels.

Several other large-scale studies have explored the relationship between RHR and fitness. The Copenhagen Male Study found an inverse association between fitness assessed submaximally in 1970 and supine RHR measured by 12-lead ECG about 15 years later in 2798 men [5]; the RHR-to-fitness relationship (beta coefficient $\cong$ -0.19 ml $O_2 \cdot kg^{-1} \cdot beat^{-1}$) was shallower than values reported in the present study at around -0.30 ml $O_2 \cdot kg^{-1} \cdot beat^{-1}$. In the Danish Health Examination Survey from 2007–2008, the relationship between seated RHR and fitness was less pronounced (beta coefficient $\cong$ -0.12 ml $O_2 \cdot kg^{-1} \cdot beat^{-1}$) when assessed with maximal cycle ergometry in over 10 thousand men and women [6]. The relationship was more pronounced in the UK Biobank study (beta coefficient $\cong$ -0.28 ml $O_2 \cdot kg^{-1} \cdot beat^{-1}$) which used an individualised submaximal cycle ergometry test to assess fitness in approximately 80,000 participants [7]. The individualisation process used RHR, however, which may limit the external validity of this association due to collinearity. A weak prospective inverse association between RHR at baseline and fitness at 23-year follow-up was reported in the Norwegian Nord-Trøndelag Health Study: -0.9 and -0.4 ml $O_2 \cdot kg^{-1} \cdot beat^{-1}$ in 807 men and 810 women, respectively [10]. The same study found within-person change in RHR between baseline and follow-up was inversely associated with fitness at follow-up; within-person change in $VO_2$max was not assessed. Among 56 thousand American patients with underlying health conditions, the age-adjusted coefficient from meta-regression across RHR categories was -0.22 ml $O_2 \cdot kg^{-1} \cdot beat^{-1}$ [8], again similar to results reported in our present study despite the difference in population sampling.

Our study is among the few to examine the influence of factors underpinning the RHR-to-fitness relationship, reporting significant inverse associations between RHR and fitness that are independent of age, sex, adiposity, and physical activity. The age, sex, BMI-, and physical activity-adjusted coefficient for seated RHR was about -0.13 ml $O_2 \cdot kg^{-1} \cdot beat^{-1}$. By comparison,

a pooled cohort analysis of almost 50 thousand American and British individuals found a similarly adjusted RHR coefficient of about -0.17 ml $O_2 \cdot kg^{-1} \cdot beat^{-1}$ [27]. However, physical activity was self-reported in those studies which may have inflated the value of the observed coefficient for RHR because of only partial adjustment for physical activity. In parallel, the Tromsø study compared seated RHR and fitness levels in 5,017 men and 5,607 women when stratified by self-reported activity levels [28], demonstrating significant inverse associations within sex and across physical activity strata. Together, these findings support the notion that RHR and habitual physical activity levels are intrinsically linked to exercise capacity. This notion is supported by previous studies that have used factors such as RHR and physical activity as well as other lifestyle factors to develop non-exercise estimation equations for fitness [29,30]. Thus, changes in fitness achieved through altered physical activity levels could be feasibly monitored with periodic RHR measurements.

RHR is associated with heart disease [31], diabetes [32], cancer [33], and all-cause, cardiovascular- and cancer-specific mortality [34,35] but the mechanisms underlying these are not fully understood. Knowing that both higher fitness and higher habitual physical activity levels are associated with lower incidence of related diseases and mortality [4], the association of RHR with fitness and the degree to which that association is influenced by physical activity and BMI explains some of the association of RHR with these endpoints. For example, a 10-year increase in RHR by 10 or 20 bpm has been shown to be associated with 18% or 31% higher all-cause mortality, respectively [36]. Applying the RHR-to-fitness beta coefficients derived in our study, these RHR values equate to declines in fitness by 2.6 and 5.2 ml $O_2 \cdot min^{-1} \cdot kg^{-1}$, respectively. Such fitness declines measured with maximal exercise testing were associated with 28% or 63% higher all-cause mortality in a small Finnish study [37] suggesting the RHR-based approximation of the dose-response relationship is attenuated.

As an example of an application of this work relevant to population monitoring, we used a smartphone application to capture weekly remote measures of RHR to examine changes in RHR by fitness level over the course of the COVID-19 pandemic. This showed that participants with lower fitness had on average a higher initial RHR and a progressively increasing RHR over time when compared to those with better fitness. This may reflect the non-uniform impact of national lockdowns on reduced physical activity opportunities [38]. Other studies that monitored RHR during the COVID-19 pandemic suggest that changes to RHR during this period may reflect disturbances in sleep patterns [39,40]; it is possible other factors may also contribute [41,42]. We could not directly measure fitness after the study observation period for reasons due to the pandemic. Therefore, future research is needed to clarify whether RHR trends in those with low fitness have lasting impact on exercise capacity.

Our study has some limitations. We used heart response to a submaximal treadmill test to estimate rather than directly measure fitness as $VO_2max$. Even though we have validated this approach [18], associations between RHR and fitness reported here may be influenced by residual error from the $VO_2max$ estimation process, which is dependent on reaching a percentage of age-predicted maximal heart rate. The standard deviation for error between predicted and directly-measured maximal heart rate is roughly 13bpm [43]. Thus, RHR-to-fitness associations reported here would likely be more precise if $VO_2max$ were directly measured. Reassuringly, associations were similar in the small subsample with direct $VO_2max$ measures. Fitness estimated from heart rate response to submaximal exercise is less reliable in those taking medications such as beta-blockers. We excluded participants on beta-blockers, as well as participants not passing the medical screening for treadmill testing, therefore our results are unlikely to generalise to these individuals. For RHR measurements during the COVD-19 pandemic, we did not directly validate the specific smartphone application that was used. Nevertheless, the general approach of using a smartphone camera to measure RHR has been

validated previously in other work [15], and the RHR values we report here by fitness level do confirm the inverse association observed using clinical measures. It is noted, however, that this approach may have lower validity in individuals with darker skin, although the majority of participants in this study were white. Finally, since the Fenland Study is specific to adult participants residing in the UK, we are unable to report on the relationship between RHR and fitness in children and adolescents, and results may also not generalise to other adult populations living in other countries.

In a population sample of UK adults, we have shown that RHR is inversely associated with fitness across different RHR measurement approaches. Half of this association is explained by modifiable factors such as body size and habitual physical activity. We also showed that within-person change in RHR was associated with within-person change in fitness and that these changes can be feasibly measured remotely, suggesting that changes in RHR may be used to track changes in fitness over time. These findings position RHR as a population-level biomarker of fitness in epidemiological and public-health settings.

## Supporting information

**S1 Checklist. STROBE statement—checklist of items that should be included in reports of** *cohort studies.*
(DOCX)

**S1 File. Supporting information–contains all the supporting tables and figures.**
(DOCX)

## Acknowledgments

We are grateful to all Fenland Study participants who gave their time and effort. We also thank the functional teams of the MRC Epidemiology Unit at Cambridge (Field Epidemiology, Study Coordination, Data management and IT) for supporting this study. We acknowledge Huma Therapeutics Ltd for their collaboration on the Fenland COVID-19 substudy, including designing and testing the smartphone application used to measure RHR.

## Author Contributions

**Conceptualization:** Tomas I. Gonzales, Justin Y. Jeon, Timothy Lindsay, Soren Brage.

**Data curation:** Tomas I. Gonzales, Kate Westgate, Stefanie Hollidge, Soren Brage.

**Formal analysis:** Tomas I. Gonzales, Timothy Lindsay, Soren Brage.

**Investigation:** Nita Forouhi, Simon Griffin, Nick Wareham, Soren Brage.

**Project administration:** Nita Forouhi, Simon Griffin, Nick Wareham, Soren Brage.

**Writing – original draft:** Tomas I. Gonzales, Justin Y. Jeon, Timothy Lindsay, Soren Brage.

**Writing – review & editing:** Tomas I. Gonzales, Justin Y. Jeon, Timothy Lindsay, Kate Westgate, Ignacio Perez-Pozuelo, Stefanie Hollidge, Katrien Wijndaele, Kirsten Rennie, Nita Forouhi, Simon Griffin, Nick Wareham, Soren Brage.

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
