## [Decision Letter · Decision Letter 0]

2 Nov 2022

PONE-D-22-26874Resting heart rate is a population-level biomarker of cardiorespiratory fitness: The Fenland StudyPLOS ONE

Dear Dr. Brage,

Thank you for submitting your manuscript to PLOS ONE. After careful consideration, we feel that it has merit but does not fully meet PLOS ONE’s publication criteria as it currently stands. Therefore, we invite you to submit a revised version of the manuscript that addresses the points raised during the review process.

We look forward to receiving your revised manuscript.

Kind regards,

Yosuke Yamada

Academic Editor

PLOS ONE

Journal Requirements:

Reviewers' comments:

Reviewer's Responses to Questions

**Comments to the Author**

1. Is the manuscript technically sound, and do the data support the conclusions?

Reviewer #1: Partly

Reviewer #2: Yes

2. Has the statistical analysis been performed appropriately and rigorously? 

Reviewer #1: Yes

Reviewer #2: Yes

3. Have the authors made all data underlying the findings in their manuscript fully available?

Reviewer #1: Yes

Reviewer #2: Yes

4. Is the manuscript presented in an intelligible fashion and written in standard English?

Reviewer #1: No

Reviewer #2: Yes

5. Review Comments to the Author

Reviewer #1: This study has a challenge to provide the information that RHR is a valid population-level biomarker of cardiorespiratory fitness. As mentioned by the authors, the measurement of fitness is hard to conduct routinely in public health settings and daily lives. Given these backgrounds, the finding of this study may contribute to the relevant area. However, the reviewer could not catch the motivation of this study, and the consistency of manuscript is not sufficient. Therefore, it seems to be difficult for future readers to follow your study. You can address some comments as follows.

Major comments

(1) The authors pointed out that the methodological reasons may result in a limited use of RHR as a population-level biomarker of fitness. Given this, how are the results of this study useful to address this problem? Throughout the study, there is no hypothesis or authors’ expectation.

First, this study measured RHR while seated, supine, and during sleep. What is the merit to compare these three ways? What results did you expect? Given that seated and supine RHR were measured under an overnight fast condition, which is not a daily life, sleeping heart rate may be suitable for daily use. What does this comparison resolve for methodological problems?

Second, what is the merit to examine the influence of modifiable factors on the RHR-to-fitness association? Please provide a reason why quantifying the influence on the RHR-to-fitness relationship strengthen the argument for using RHR as a population-level biomarker of fitness. There is a possibility that the criterion-related validity of RHR to fitness may improve when a factor affecting the association between RHR and fitness is considered. The relevant discussion (L301, 389-390) should be also modified with this viewpoint.

Third, it is very hard to catch the role of data during the pandemic in this study (Figure 3 and Supplementary Figure 2 and 3). The relevant discussion (L294-299) should be also modified with this viewpoint. If necessary, it is better to delete this part in this paper.

(2) Related to the above-mentioned comment, the information of previous study in this area was limited in Introduction. Therefore, it is hard for future readers to follow the gap between this work and previous works. Why did not the authors cite the literatures (ref. 23, 24, 25) in Introduction? What is the difference between these studies and this study? Please explain.

(3) For the main results in this study, all of data including fitness were assessed based on heart rate. This may lead to overestimate the association between RHR and fitness. Please further discuss this point.

Minor comments

Abstract

It is better to provide that the population during the coronavirus disease-2019 pandemic is also a sub-group of the UK Fenland Study.

L39-40

Where are these results from? Please specify. (If these results are based on the cross-sectional analysis, why sex-adjusted?)

L45-46

Is this a conclusion based on your findings? It is better to refer to your results.

Introduction

L73-75

Please provide a reason why RHR could a viable alternative. Is RHR correlated with VO2max? How much? Is RHR also associated with the lower risk of diabetes, CVD, cancer, and mortality?

Methods

L102-103

Participants with diabetes were excluded. Why did the study exclude only diabetes? How about other chronic diseases such as hypertension and dyslipidemia?

Moreover, what does terminal illness include?

Supplementary Table 1

It is better to provide each effect size instead of P value, because the number of participants is large.

L109-L124

It is better to provide the flowchart of selection of participants, because the study was conducted in the same study population (the UK Fenland Study). The STROBE statement recommends the use of a flow diagram.

L134-138

Please provide the details (company, model number, etc.) of sensor measuring sleep heart rate.

L214-216d

Why did the authors exclude the missing continuous variables? It is better to use multiple imputation method.

Results

L232-233

Is RHR is supine position? Please specify.

Supplementary Table 3

This table is very confusing. Do the results represent men and women? Why are regression equations needed? Isn't a correlation matrix sufficient?

Figure 1

Is “R” in the figure caption equal to “r” in each figure? Please match.

Reviewer #2: The authors provide useful information for RHR as a biomarker. Particularly they provide findings on seated, supine, and sleeping heart rate.

These findings support the use of non-exercise fitness equations eCRF that use RHR as a key algorithm parameter. See, Wang Y, Chen S, Lavie CJ, Zhang J, Sui X. An overview of non-exercise estimated cardiorespiratory fitness: estimation equations, cross-validation and application. J Sci Sport Exerc 2019 Jun 10;1(1):38-53.

There are studies that provide correlations information between MAX VO2 and RHR.

See Sloan R, Visentini-Scarzanella M, Sawada S, Sui X, Myers J, Estimating Cardiorespiratory Fitness Without Exercise Testing or Physical Activity Status in Healthy Adults: Regression Model Development and Validation, JMIR Public Health Surveill 2022;8(7):e34717

Artero et al 2014 showed that eCRF was a better predictor than of all cause mortality than RHR alone. RHR is more useful when applied to eCRF equations.

6. PLOS authors have the option to publish the peer review history of their article (what does this mean?). If published, this will include your full peer review and any attached files.

Reviewer #1: No

Reviewer #2: No

---

## [Decision Letter · Decision Letter 1]

3 Apr 2023

PONE-D-22-26874R1Resting heart rate is a population-level biomarker of cardiorespiratory fitness: The Fenland StudyPLOS ONE

Dear Dr. Brage,

Thank you for submitting your manuscript to PLOS ONE. After careful consideration, we feel that it has merit but does not fully meet PLOS ONE’s publication criteria as it currently stands. Therefore, we invite you to submit a revised version of the manuscript that addresses the points raised during the review process.

We look forward to receiving your revised manuscript.

Kind regards,

Yosuke Yamada

Academic Editor

PLOS ONE

Journal Requirements:

Reviewers' comments:

Reviewer's Responses to Questions

**Comments to the Author**

1. If the authors have adequately addressed your comments raised in a previous round of review and you feel that this manuscript is now acceptable for publication, you may indicate that here to bypass the “Comments to the Author” section, enter your conflict of interest statement in the “Confidential to Editor” section, and submit your "Accept" recommendation.

Reviewer #1: (No Response)

Reviewer #2: All comments have been addressed

Reviewer #3: (No Response)

2. Is the manuscript technically sound, and do the data support the conclusions?

Reviewer #1: Partly

Reviewer #2: Yes

Reviewer #3: Yes

3. Has the statistical analysis been performed appropriately and rigorously? 

Reviewer #1: Yes

Reviewer #2: Yes

Reviewer #3: Yes

4. Have the authors made all data underlying the findings in their manuscript fully available?

Reviewer #1: Yes

Reviewer #2: Yes

Reviewer #3: Yes

5. Is the manuscript presented in an intelligible fashion and written in standard English?

Reviewer #1: Yes

Reviewer #2: Yes

Reviewer #3: Yes

6. Review Comments to the Author

Reviewer #1: (No Response)

Reviewer #2: I have no further comments to be made. 

Reviewer #3: The reviewers comments have been addressed well.

Minor Comment:

Methods

The methods are clearly and thoroughly explained. However, in lines 164-165, I would like to see more detail regarding the treadmill protocol that was used. Currently, the protocol is very general in its description. Was this a standard treadmill protocol, i.e. Bruce test?

7. PLOS authors have the option to publish the peer review history of their article (what does this mean?). If published, this will include your full peer review and any attached files.

Reviewer #1: No

Reviewer #2: **Yes: **Robert A Sloan

Reviewer #3: No

---

## [Author Response · Author response to Decision Letter 1]

11 Apr 2023

Reviewer #3: The reviewers comments have been addressed well.

Minor Comment:

Methods

The methods are clearly and thoroughly explained. However, in lines 164-165, I would like to see more detail regarding the treadmill protocol that was used. Currently, the protocol is very general in its description. Was this a standard treadmill protocol, i.e. Bruce test?

RESPONSE

We now provide additional text (in red) in the present manuscript to detail the treadmill protocol used – this information was available in the cited reference but we agree that it is more reader friendly to include the extra detail here. We now cite the previous work earlier in this paragraph and we have also provided an extra citation to a recently published article that described the validity and epidemiological utility of the fitness estimates arising from this protocol.

---

## [Editor Report · Decision Letter 2]

19 Apr 2023

Resting heart rate is a population-level biomarker of cardiorespiratory fitness: The Fenland Study

PONE-D-22-26874R2

Dear Dr. Brage,

We’re pleased to inform you that your manuscript has been judged scientifically suitable for publication and will be formally accepted for publication once it meets all outstanding technical requirements.

Kind regards,

Yosuke Yamada

Academic Editor

PLOS ONE
---

## [Editor Report · Acceptance letter]

2 May 2023

PONE-D-22-26874R2 

Resting heart rate is a population-level biomarker of cardiorespiratory fitness: The Fenland Study. 

Dear Dr. Brage:

I'm pleased to inform you that your manuscript has been deemed suitable for publication in PLOS ONE. Congratulations! Your manuscript is now with our production department. 

Kind regards, 

on behalf of

Dr. Yosuke Yamada 

Academic Editor

PLOS ONE